# Effect of Foliar Spray Application of Zinc Oxide Nanoparticles on Quantitative, Nutritional, and Physiological Parameters of Foxtail Millet (*Setaria italica* L.) under Field Conditions

**DOI:** 10.3390/nano9111559

**Published:** 2019-11-03

**Authors:** Marek Kolenčík, Dávid Ernst, Matej Komár, Martin Urík, Martin Šebesta, Edmud Dobročka, Ivan Černý, Ramakanth Illa, Raghavendra Kanike, Yu Qian, Huan Feng, Denisa Orlová, Gabriela Kratošová

**Affiliations:** 1Department of Soil Science and Geology, Faculty of Agrobiology and Food Resources, Slovak University of Agriculture in Nitra, Tr. A. Hlinku 2, 949 76 Nitra, Slovakia; 2Nanotechnology Centre, VŠB Technical University of Ostrava, 17. listopadu 15/2172, 708 00 Ostrava-Poruba, Czech Republic; denisa.orlova.st@vsb.cz (D.O.); Gabriela.Kratosova@vsb.cz (G.K.); 3Department of Crop Production and Grassland Ecosystems, Faculty of Agrobiology and Food Resources, Slovak University of Agriculture in Nitra, Tr. A. Hlinku 2, 949 76 Nitra, Slovakia; david.ernst@uniag.sk (D.E.); xkomarm@uniag.sk (M.K.); ivan.cerny@uniag.sk (I.Č.); 4Institute of Laboratory Research on Geomaterials, Faculty of Natural Sciences, Comenius University in Bratislava, Ilkovičova 3278/6, Karlova Ves, 841 04 Bratislava, Slovakia; urik@fns.uniba.sk (M.U.); martin.sebesta@uniba.sk (M.Š.); 5Institute of Electrical Engineering, Slovak Academy of Sciences, Dúbravská cesta 9, 841 04 Bratislava, Slovakia; edmund.dobrocka@savba.sk; 6Department of Chemistry, Rajiv Gandhi University of Knowledge Technologies, AP IIIT, Nuzvid, Krishna District 521202, India; ramakanthilla@yahoo.com; 7Department of Biosciences, Rajiv Gandhi University of Knowledge Technologies, AP IIIT, Nuzvid, Krishna District 521202, India; raghavendrakanike@gmail.com; 8School of Ecology and Environmental Science, Yunnan University, 2 Cuihubei Lu, Kunming 650 091, Yunnan, China; qianyu@ynu.edu.cn; 9Department of Earth and Environmental Studies, Montclair State University, 1 Normal Ave, Montclair, NJ 070 43, USA; fengh@montclair.edu

**Keywords:** foliar application, zinc-oxide nanoparticles, foxtail millet, quantitative, nutritional, and physiological parameters, nano-fertilizers

## Abstract

It has been shown that the foliar application of inorganic nano-materials on cereal plants during their growth cycle enhances the rate of plant productivity by providing a micro-nutrient source. We therefore studied the effects of foliarly applied ZnO nanoparticles (ZnO NPs) on *Setaria italica* L. foxtail millet’s quantitative, nutritional, and physiological parameters. Scanning electron microscopy showed that the ZnO NPs have an average particle size under 20 nm and dominant spherically shaped morphology. Energy dispersive X-ray spectrometry then confirmed ZnO NP homogeneity, and X-ray diffraction verified their high crystalline and wurtzite-structure symmetry. Although plant height, thousand grain weight, and grain yield quantitative parameters did not differ statistically between ZnO NP-treated and untreated plants, the ZnO NP-treated plant grains had significantly higher oil and total nitrogen contents and significantly lower crop water stress index (CWSI). This highlights that the slow-releasing nano-fertilizer improves plant physiological properties and various grain nutritional parameters, and its application is therefore especially beneficial for progressive nanomaterial-based industries.

## 1. Introduction

Cereals are one of the most important global commercial crops [1], and this applies equally to the Slovakian agriculture [2]. One widely used cereal is the *Setaria italica* L. foxtail millet, which has a higher nutritional value than rice [3]. In addition to its production as a food and fodder crop in several parts of Eastern Europe, Asia, and Africa [4,5], foxtail millet has been shown to prevent soil erosion [6] and to have a positive impact on biodiversity [7]. The positive effects of metal (metal oxide) nanoparticles (NPs) on crops have recently been recorded, and these include being an effective growth stimulator in dry seasons, enhancing the yield and nutritional quality of agricultural products [8,9,10], and providing effective mediation as an active anti-pathogen agent [11,12,13].

Zinc has a major role as a biological micronutrient in plant systems, and this includes its important functions in metal-protein complexes [14]. Further zinc functions operating within a specific concentration range include its activity in plant protection, formation of a stabilizing agent for cell membranes, support for protein synthesis, wide-spread membrane functions for cell elongation, and maintaining vital protection against environmental stress [14,15,16,17].

In addition, Nandhini et al. (2019) confirmed ZnO NPs’ positive effect on pearl millet germination and growth, and Singh et al. (2019) and Rizwan et al. (2019a) demonstrated the positive effects of ZnO NPs on the quantitative, nutritional, and physiological parameters of *Triticum aestivum* L. wheat and *Zea mays* L. maize. These also included the contents of starch, oil, and total proteins [11,18,19]. The ZnO NPs further enhanced the activity of pearl millet’s defense enzymes, such as phenylalanine ammonia-lyase, lipoxygenase, polyphenol oxidase, and peroxidase in resistance to plant fungal attack [20] and exerted a positive effect on chlorophyll synthesis in both corn and wheat plants [21,22].

Characterization of the physical properties of zinc-containing particles is necessary when they are not in free or complexed-ion form, and this is especially important when considering the size, morphology, crystallinity, and stability of nanoparticles [8,14,23]. While foliar-applied ZnO NP concentrations can have both positive and negative effects on plant growth and development compared to the ionic form (Zn^2+^) of corresponding metal and micro(macro)particles [17], most of the reported results of ZnO NPs’ impact on crops are based on laboratory experiments and hydroponic cultures. Moreover, field experiments are quite rare and reports on direct foliar application of ZnO NPs are mostly lacking in recent publications. This therefore inspired us to conduct field experiments over one season to evaluate the precise effects of foliarly applied ZnO NPs on foxtail millet’s quantitative, nutritional, and physiological parameters under field conditions.

## 2. Materials and Methods

### 2.1. Characterization of ZnO Nanoparticles

The zinc oxide nanoparticles for foliar application were obtained from Sigma-Aldrich (Saint-Louis, MO, USA). These 721077commercial code ZnO NPs with ≤40 nm average particle size, 20 wt. %, were diluted in H_2_O and then analyzed by X-ray diffraction (XRD) by Bruker D8 DISCOVER diffractometer (Bruker, MA, USA). A 12 kW (40 kV, 300 mA) Cu-anode was employed for XRD measurements, and unit cell parameters were calculated in TOPAS 3.0 software (Burker, MA, USA). JEOL 7610 F+ scanning electron microscopy (SEM) visualized the nanoparticles’ dimension range and morphology, and the energy dispersive X-ray spectrometer EDAX on Phillips XL30 verified the chemical composition of the qualitative components.

### 2.2. Site Description and Field Experiments

The field experiments were conducted at Hostie village in the Slovak Republic. This is situated at 310 m above sea level in the northeastern portion of the Žitavská upland area at the foot of the Tríbeč mountains in the Hostianský riverbed valley (Figure 1). Geomorphological classification identifies the experimental locality in upland-like areas as ilimerized and brown forest soils [24] where intensive soil-cultivation enables annual potato and maize rotation.

The experiments were performed in 5 m^2^ micro-parcels with three replications of each treatment [25,26]. Two experimental variants were established by randomly arranged, perpendicularly divided blocks (Figure 2) and a trail without foliate-applied zinc oxide nanoparticles formed the control.

The experimental area was used for cultivating potatoes in the season before our experiment and the conventional procedure of deep ploughing was then performed [27,28]. The pH, carbonates content [29], grain-size distribution [30,31], concentrations of micronutrients [32], and Zn availability in soil were measured using DTPA extraction [33] before tillage (Table 1) and fertilizers were then applied manually.

The initial solutions of K_2_SO_4_, with 50% K_2_O and superphosphate with 17% P_2_O_5_ concentration from AGRO CS Slovakia a. s. (Veľké Dravce, Slovak Republic), were implemented during autumn. LAD 27 was then applied after spring for pre-sowing soil treatment; LAD 27 is a nitrogen-type fertilizer containing 27% nitrogen, 4.1% MgO, 7% total CaO, and 2% water-soluble CaO (Duslo, a. s., Šaľa, Slovak Republic).

The experimental foxtail millet was manually sown in lines with 20 mm sowing depth, 100 mm seeding distance, and 200 mm inter-row spacing, and the plots were pressed by roller [27]. The ZnO NPs were then suspended in a solution of adjuvant SILWET STAR^®^ before foliar application because this facilitates penetration of nanoparticles across wax sub-structures [34,35] and also used for control as a “blank” treatment lacking ZnO NPs.

The final zinc concentration in suspension was 2.6 mg.L^−1^ and the foliate spray was applied twice during the growth season when foxtail millet reached ideal phenological growth phases on the 27th and 53rd days of the vegetation period recommended by Meier (1997) [36] (Figure 3). The plants were then sprayed with GAMMA 5 (Mythos Di Martino, Mussolente, Italy) on an early and windless morning until all plants were completely wet.

### 2.3. Analysis of Foxtail Millet Quantitative and Nutritional Parameters

All plants were harvested manually when they reached full grain maturity, and the required 11% moisture was tested by HE Lite (Pfeuffer GmbH, Kitzingen, Germany). The following quantitative plant parameters were established: the number of plants and seed heads were counted manually, plant height and seed head length were determined in millimeters by SOLA VF 3m laboratory equipment (SOLA-Messwerkzeuge GmbH, Götzis, Austria), dry seed head and grain yield gram-weights were measured on the KERN PCB3500-2 laboratory scale (KERN & Sohn GmbH, Balingen, Germany), and the gram-weight of the thousand grains (TGW) was recorded by NUMIREX equipment (MEZOS spol. s r.o., Hradec Králové, Czech Republic).

In addition, the following percentages were also established: the oil content in percent was determined using the SOXSHLET method [37], starch content was obtained by the Ewers polarimetric method [38], and the dry mass content was established gravimetrically after samples were dried at 105 °C to achieve constant weight. Finally, the total protein percentage of nitrogen contained in organic compounds and in ammonia and ammonium in inorganic compounds, as well as the total nitrogen concentration in mg.kg^−1^, were determined after the mineralization process with H_2_SO_4_ as in Kjeldahl [32].

### 2.4. Observation of the Foxtail Millet Crop Water Stress Index Physiological Parameter

The primary method used in crop water stress index (CWSI) detection is to measure and assess the leaf temperature (Tc), air temperature (Ta), and atmospheric vapor pressure [39,40,41]. These results were then applied in Equation (1):CWSI = (T_c_ − T_a_) − (T_c_ − T_a_)_u_/(T_c_ − T_a_)_ul_ − (T_c_ − T_a_)_u_(1)
where (Tc − Ta) corresponds to the actual temperature difference in °C; (Tc − Ta)_u_ indicates the virtual temperature differences in large water supply by sensing the green-paper moisture; and (Tc − Ta)_ul_ reveals the differences in leaf-desiccation sensed by the yellow-brown paper [42].

The non-destructive infrared thermal method analyzed the following physiological parameters: the leaf temperature was sensed by BASETech IRT-350 (Conrad Electronic SE, Hirschau, Germany), and air temperature was monitored by TFA 12.2046.61 (TFA Dostmann GmbH & Co. KG, Wertheim-Reicholzheim, Germany). A physiologically mature foxtail millet plant leaf was chosen during the growing season, and the plants were then labelled and measured from 11 am to 1 pm on the specific days listed in Table 2. This table lists the five infrared thermal measurements performed in the experiment, and these are also illustrated in Figure 3.

### 2.5. Statistical Analysis

All experimental results were evaluated by standard statistical and graphic methods in Statistica 10 software (StatSoft, Inc., Tulsa, OK, USA) and then subjected to analysis of variance (ANOVA) and Fisher’s least significant difference (LSD) test at α = 0.05 significance.

## 3. Results and Discussion

### 3.1. Zinc Oxide Nanoparticle Characterization

Scanning electron microscopy revealed that the ZnO NPs applied in the field experiments were spherical, columnar, hexagonal rod-like, or almost cuboidal in shape (Figure 4a). Their mean size is approximately 17.3 ± 0.1 nm (Table 3).

Electron dispersive analysis (EDAX) proved that the major particles’ components were zinc and oxygen (Figure 4a), and X-ray diffraction analysis highlighted the ZnO NPs’ wurtzite-type structural characteristics (Figure 4b). The unit cell parameters are listed in Table 3.

Despite the ZnO NPs’ size and shape in the solution applied in the field experiment, it is quite likely that ZnO NPs undergo gradual photo-corrosion when exposed to sunlight on the leaf surfaces [43], and therefore, the nanoparticle input to plants from leaves and the related physiological effects are a little different to those in the zinc-based ionic form Zn^(2+)^ and micro-particle foliarly applied fertilizers [17].

### 3.2. Effects of Zinc Oxide Nanoparticles on Foxtail Millet’s Quantitative, Nutritional, and Physiological Parameters

Table 4 highlights that there were no statistically significant differences in plant height and seed head length between plants treated with ZnO NPs and the control.

No differences were established between the weights of dry seed head and thousand grains (TGW) of ZnO NPs foliarly applied treatments and the control; however, this was expected because the concentration of zinc was significantly lower than the reported 0.1% value that usually has harmful effects when ionic form of Zn^(2+)^-corresponding metal-based fertilizer is applied foliarly [44].

In addition, although the positive effect of the approximately 62 mg/L^−1^ ZnO NPs concentration on growth parameters was confirmed by Singh (2019), and a higher concentration effect was indicated by [8] or [45]. The specific dose-dependency of foliarly applied ZnO NPs appears more complex when linked with factors such as size, shape, surface-to-volume ratio [23], seasonal period, and the number of applications. There is, however, no available information in the literature on the effect of less than 3 mg.L^−^ ZnO NPs on plant production and nutritional parameters under field conditions.

Similar to other authors’ reports on various crops [21,46], ZnO NP foliar application affected various foxtail millet nutritional parameters, and our results indicate a statistically significant increase in oil content following ZnO NP application (Table 4). The ZnO NP-treated foxtail millet seed oil content was 12.5%, while the oil content in the control treatment was lower by 25%. This result has great importance for agricultural science because the 1995 FAO report recorded an average content of approximately 4% oil in foxtail millet grains and an 88.4% average dry mass content. The dry mass content there also registered almost identical values for both the ZnO NP and control treatments, but experimental values prove higher than those in the FAO reference [47].

The higher registered oil production values may be related to ZnO NPs’ ability to stimulate enzymatic activity, as this has also been recorded for pearl millet in greenhouse conditions [20]. Further, Rizwan et al. (2019a; 2019b) suggested that ZnO NP-treated maize and wheat had higher chlorophyll concentrations, and they indicated that the resultant greater photosynthetic effectiveness can be directly linked to the total protein, starch, oil, and dry mass contents [11,18,19].

In contrast, although experimental total protein contents were not statistically different in the control and ZnO NP-treated plants, we identified a statistically significant decrease in starch content in the control plants (Table 4). This is not surprising because some reports have indicated a negative correlation between starch and oil content [48], and other authors suggest that a significantly higher concentration of ZnO NPs is required to induce starch accumulation. This was also previously reported in *Solanum tuberosum* L. potatoes, where the application of 100, 350, and 500 mg.L^−1^ ZnO NPs increased the average starch content by 100% [49]. In addition, there is interesting research showing that the photocatalytic effect occurring during early interaction of ZnO NPs with the leaf surface slightly accelerates the photosynthesis rate [22].

This higher photosynthesis rate in plants commonly reduces crop water stress (CWSI) [50,51], and this strongly supports our results, where all measurements taken after ZnO NP foliar application confirmed the lowest CWSI values compared to the control (Table 4, Figure 5). Moreover, Kirnak et al. (2019) also highlighted that the oil content in pumpkin seed increased with decreasing CWSI.

## 4. Conclusions

Herein, we evaluated the effects of foliarly applied ZnO NPs on the quantitative and nutritional characteristics of foxtail millet grains. Concentrations of 2.6 mg.L^−1^ ZnO NPs with an average particle size under 20 nm were twice applied foliarly in our field experiment, and the most significant differences in millet grain oil and the total nitrogen content nutritional parameters were recorded for plants treated with ZnO NPs and untreated controls. While we noted that the experimental grain dry matter and starch content were comparable to untreated plants and no significant differences were detected in the plant height after ZnO NPs, seed head length, and thousand-grain weight quantitative parameters, differences were registered in nutritional parameters. In addition, the plant water stress index was lower after ZnO NP foliar application than in the control throughout the entire life cycle. Moreover, the increased foxtail millet grain nutritional parameters highlighted improved plant photosynthetic efficiency, transpiration, and enzymatic activity. Finally, these results confirm our conclusion that the foliar application of ZnO NP-based fertilizers improves the quality of crop products, and this provides a most efficient strategy for future agricultural industry management in dry areas where this crop prevails.

## Figures and Tables

**Figure 1 nanomaterials-09-01559-f001:**
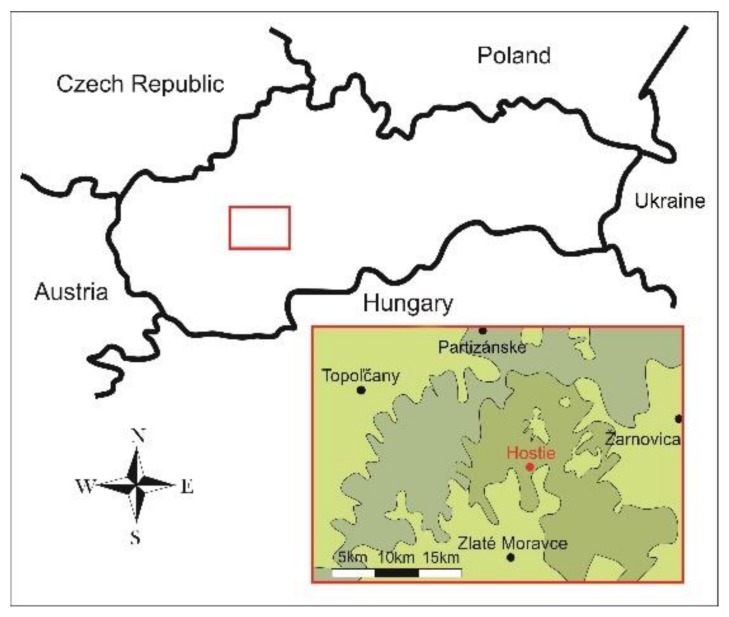
Schematic illustration of the Hostie experimental locality near Zlaté Moravce in the Slovak Republic. Nitrianska pahorkatina—Hills and the Tribeč mountain region are in the western part, the Pohronský Inovec mountain is in the east, and the Žitavská upland area is in the southern portion [24].

**Figure 2 nanomaterials-09-01559-f002:**
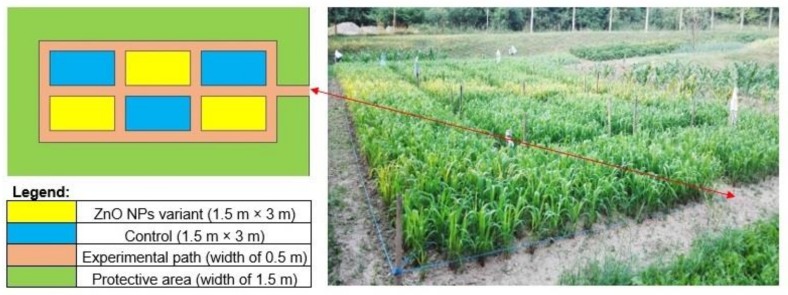
Experimental plan visualization.

**Figure 3 nanomaterials-09-01559-f003:**
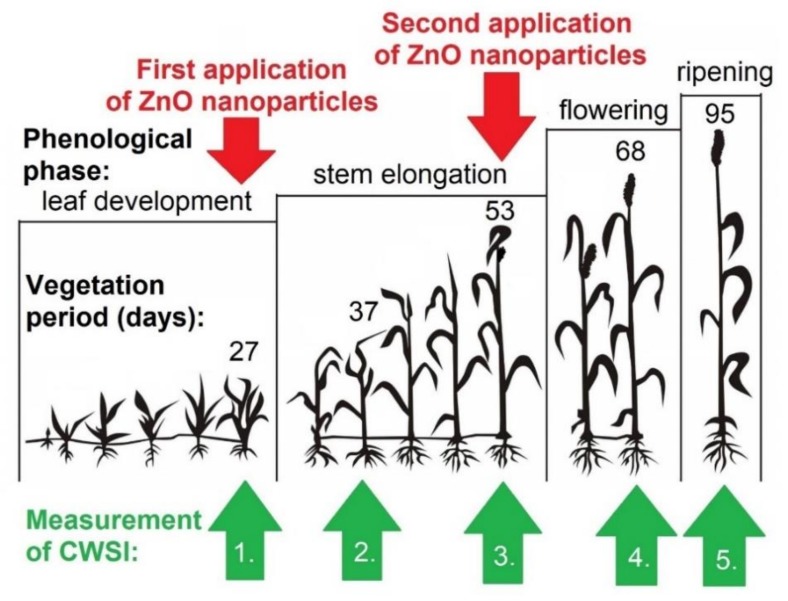
Schematic illustration of the crop phenological phases; the red arrows indicate the two vegetation periods at 27 and 53 days of foliar spray application with 2.6 mg.L^−1^ zinc oxide nanoparticle concentration. The green arrows record the crop water stress index refraction measurements (CWSI).

**Figure 4 nanomaterials-09-01559-f004:**
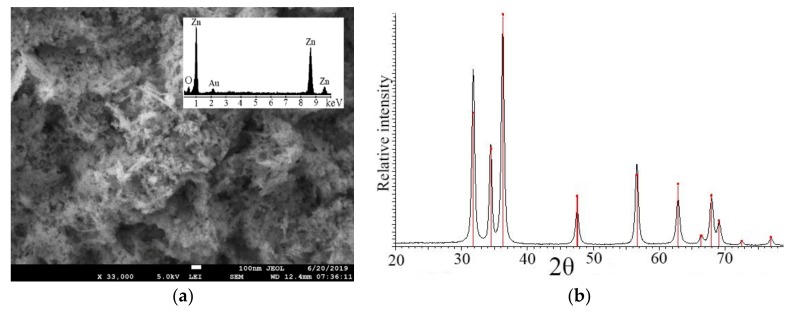
(**a**) Scanning electron microscopy (SEM) shows zinc oxide nanoparticles forms (ZnO NPs) foliarly applied to foxtail millet. Inset: The electron dispersive analysis (EDAX) profile where Zn and O are detected as the major elements. (**b**) X-ray diffraction analysis verified that the zinc oxide nanoparticles have wurtzite-structural symmetry.

**Figure 5 nanomaterials-09-01559-f005:**
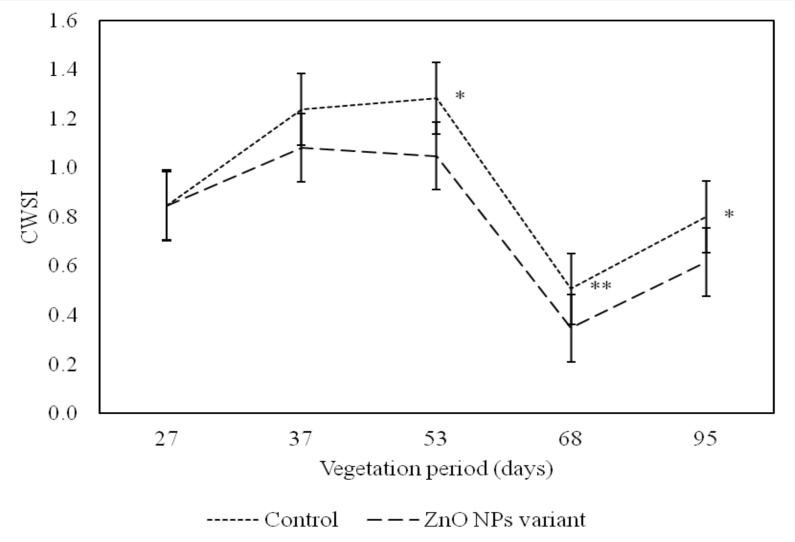
The effect of foliarly applied ZnO NPs vs. control (without ZnO nanoparticle application) related to foxtail millet crop water stress index (CWSI) measured during the 2018 growth season. Significances are: * *P* value < 0.05, ** *P* value < 0.01.

**Table 1 nanomaterials-09-01559-t001:** Selective soil characteristics from the experimental locality before experimental trials.

pH	Carbonates Content (%)	Grain-Size Distribution (mm) %		Content of Nutrients (mg.kg^−1^)			
7.01	3.55 ± 0.06	up to 0.002	25.66	**N_in_**	**P**	**K**	**Zn _available_**
		0.002–0.05	63.04	23.2	82.5	200	4.24 ± 0.24
		0.05–2000	11.3				
		silt loam					

**Table 2 nanomaterials-09-01559-t002:** Measurements of foxtail millet foliate crop water stress index (CWSI) in (1) foxtail millet foliate treated with zinc oxide nanoparticles (ZnO NPs) and (2) in controls; at different growth stages.

Measurement Number	Dates	Report
1.	27 May 2018	Measurement before first ZnO NPs application
2.	6 June 2018	Measurement at two weeks after first ZnO NPs application
3.	22 June 2018	Measurement before the second ZnO NPs application
4.	7 July 2018	Measurement at two weeks after second ZnO NPs application
5.	3 August 2018	Measurement before harvest

**Table 3 nanomaterials-09-01559-t003:** X-ray diffraction analysis of ZnO NPs with calculated unit cell parameters.

Crystal Symmetry	Hexagonal
*a*-axes	3.25077 ± 0.00008 Å
*c*-axe	5.2097 ± 0.0002 Å
*α*	90°
*γ*	120°
Space group	*P*6_3_*m*
Unit cell volume	47.58 Å^3^ (Calculated from Unit Cell)
L_vol_-IB	17.3 ± 0.1 nanometer (Calculated from X-ray diffraction data)

**Table 4 nanomaterials-09-01559-t004:** Comparison of foxtail millet quantitative, nutritional, and physiological parameters in the zinc oxide nanoparticles (ZnO NPs) foliarly applied variant and the control when harvested after 95 days. This included the standard deviation (±SD) and calculation by Fisher’s least significant difference (LSD) test.

	ZnO NPs Foliarly Applied Variant	Control Variant (without ZnO NPs Application)
Quantitative parameters		
Plant high (mm)	1089 ± 121	1031 ± 192
Seed head length (mm)	79.72 ± 8	76.94 ± 17
Weigh of dry seed head (g)	2.54 ± 0.24	2.91 ± 0.33
Weight of thousand grains (TGW) (g)	5.19 ± 0.51	5.37± 0.89
Grain yield (g)	1244 ± 199	1304 ± 157
Nutritional parameters		
Content of total nitrogen N_tot_ (mg.kg^−1^)	17611 ± 38 **	17302 ± 11 **
Content of oil (%)	12.5 ± 0.29 **	9.33 ± 0.003 **
Content of starch (%)	48.23 ± 0.002 **	48.9 ± 0.002 **
Dry mass (%)	89.51 ± 0.02 *	89.42 ± 0.04 *
Total proteins (%)	11.77 ± 1.06	10.79 ± 0.05
Physiological parameter		
CWSI ^1^	0.7875 ± 0.042 **	0.9345 ± 0.031 **

^1^ Crop water stress index (CWSI), the significance: * *P* value < 0.05, ** *P* value < 0.01.

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
