# Peer review of "Effect of Foliar Spray Application of Zinc Oxide Nanoparticles on Quantitative, Nutritional, and Physiological Parameters of Foxtail Millet (Setaria italica L.) under Field Conditions"

_nanomaterials, 2019, doi:10.3390/nano9111559_

Round 1
Reviewer 1 Report
Comments to Authors
This is an interesting manuscript describing the effect of the foliar application of ZnO NPs to a cereal under field conditions. The manuscript includes a detailed methodology of the plant growth and the parameters analyzed. The manuscript matches the scope of the journal and presents interesting novel research. There are a few points that need to be clarified and reviewed.
Introduction
The introduction part is poor. In recent years, there are several works that are using the Zn NPs and the revision done by the authors can be improved. Moreover, there is a lack of description of the role of Zn in plants and while it is an essential micronutrient.
M&M
The material and Methods section is, in general, well described. However, there are a few points that need clarification:
The experiment design included the ZnNPs treatment and a control, but a comparison with a ZnO (non NPs) was not included. Why the authors did not include this treatment?
Only three replicates were assayed per treatment and the experiment did not include blocks. Under field conditions, at least 5 replicates, and two blocks are generally used. However, it is curious that the standard error is quite low in all the parameters analyzed. How was the material selected for the subsequent “qualitative parameters” analyzed? How much material was analyzed for each parameter? The authors must clarify and specify this information in lines 122-134.
Some soils characteristics are presented in Table 1. This description is quite short. What about the texture, carbonate content, and the Zn available concentration (DTPA extraction)? These parameters are important to understand if the soil was Zn deficient or not.
In lines 112-113, the abbreviation BBCH needs to be defined.
Line 133, how was the total protein content determined? Include a brief description.
Lines 165-170, this paragraph must be included in the introduction section.
Line 186: “nanoparticles’ input into plants from leaves and the related physiological effects are a bit different in comparison to soluble zinc-based fertilizers [35].” What did the authors want to say exactly when referring to soluble zinc-based fertilizers? Both Zn fertilizers applied foliar and to the soil?
Table 4. When the parameters were analyzed? At the end of the experiment, thus, 72 days? Instead of referring to “during the growing season” in the Table heading the specific day must be included.
Instead of including the P values I suggest to delete this column and include symbols of asterisks indicating the significance. (e.g *P value <0.05, **P value <0.01.)
The term “qualitative parameters” is confusing. They refers to parameters of quality, but they are quantitative (you have a mean value for each). Revise this term over the manuscript. Moreover, include the organ that was analyzed (grains).
Lines 200-201: “This was expected as the concentration of Zn was significantly lower than the reported 0.1% value that have the harmful effects when applied foliarly [36].” Include this information in the introduction section.
Lines 203-204: “Thus, the positive effect of ZnO NPs on standard growth parameters is more significant when uptaken by roots even at low concentration range [38].” This sentence is not clear. Are the authors comparing to ZnNPs applied to soil? Or Zn (non-NPs) applied to the soil?
Lines 215-216: “ZnNPs also could inhibit the pathogen manifestation, which in turn affected the quality of final crop products [11” Did the authors observe this effect in their experiment? If not, remove this dissertation.
Line 220-221: authors may refer to control plants, not to control experiments.
Figure 5. Remove “variant” from legend and figure caption. In the X ax include the days (31, 37, etc.) instead of the number of measurements to present a quantitative ax instead of the qualitative ax. Include *, **, etc. with the ANOVA analysis for each day.
Author Response
Comments and Suggestions for Authors
Reviewer I, Comment #1,
I would suggest only some language editing before publishing and some minor changes according to the following comments:
Answer: English language corrections involved in manuscript was checked by English lector Dr. Raymond Joseph Marshall.
Comment #2, Comment #8,
The introduction part is poor. In recent years, there are several works that are using the Zn NPs and the revision done by the authors can be improved. Moreover, there is a lack of description of the role of Zn in plants and while it is an essential micronutrient.
Answer: We agree, the description of zinc (ZnO NPs, Zn-soluble species, and Zn microparticles) was upgraded into manuscript in intro section with related references.
…”Zinc has a major role as a biological micronutrient in plant systems, and this includes its important functions in metal-protein complexes [1]. Further zinc functions operating within a specific concentration range include its activity in plant protection, formation of a stabilising agent for cell membranes, support for protein synthesis, wide-spread membrane functions for cell elongation and maintaining vital protection against environmental stress [1-4].”…
Added references to manuscript: Estrada-Urbina et al. 2018 [3], Wagner et al. 2016 [5], Gkanatsiou, et al. 2019 [6], Mousavi Kouhi et al. 2014 [4]
Comment #3, + Comment #8,
The experiment design included the ZnNPs treatment and a control, but a comparison with a ZnO (non NPs) was not included. Why the authors did not include this treatment?
Answer: Design of experiment was settled with ZnO nanoparticles variant (and control) based on hypothesis of different nanoparticles properties (size, morphology, surface-to-volume, and effect + others physico-chemical properties) in comparison with micro-particles, and conventional ionic form of Zn(2+) corresponding metal fertilizers. Moreover, according to suggestion “paragraph” from discussion was placed for more “strongest” description of nano-related effect to intro section + reference [4].
.… “Characterisation of the physical properties of zinc-containing particles is necessary when they are not in free or complexed-ion form and this is especially important when considering the size, morphology, crystallinity and stability of nanoparticles [1, 7, 8].”….
…”While foliar-applied ZnO NPs concentrations can have both positive and negative effects on plant growth and development compared to ionic form (Zn2+) of corresponding metal and micro(macro)particles [4],”…
Comment #4,
Only three replicates were assayed per treatment and the experiment did not include blocks. Under field conditions, at least 5 replicates, and two blocks are generally used. However, it is curious that the standard error is quite low in all the parameters analyzed. How was the material selected for the subsequent “qualitative parameters” analyzed? How much material was analyzed for each parameter? The authors must clarify and specify this information in lines 122-134.
Answer: For ANOVA representative analysis are needed to have minimum 3 replications (this point was done). Our long-term empirical evidences confirmed more appropriate undivided parcels which resulted to more plant growth and “compact” vegetation and reproducible plants development instead of (2x5), otherwise, if micro-parcels are divided into less segments effect of marginal vegetation could be most likely appear. For micro-parcel experimental design, we followed standard methodology according to [9, 10]. All plants from all un/treatment area – experimental trials (from production to nutritional parameters) were analyzed. Average number of plants: 530±70.45 pcs (ZnO NPs variant) and 490±102.78 pcs (control). Total number of plants: 3060 pcs (6 experimental trials). Average number of seed heads: 505±96.46 pcs (ZnO NPs variant) and 490±102.78 pcs (control). Total number of seed heads: 2986 pcs (6 replicates). Nutritional parameters were analyzed from a sample containing the whole grain yield of each replicate (6 samples). Average weight of 1 sample: 1244±199.04 g (ZnO NPs variant) and 1304±157.10 g (control).
...“ All plants were harvested manually when they reached full grain maturity, and the required 11% moisture was tested by HE Lite (Pfeuffer GmbH, Kitzingen, Germany).?...
Comment #5,
Some soils characteristics are presented in Table 1. This description is quite short. What about the texture, carbonate content, and the Zn available concentration (DTPA extraction)? These parameters are important to understand if the soil was Zn deficient or not.
Answer: Soil properties including carbonates content, grain-size distribution, Zn available concentration using DTPA extraction were added into manuscript (see Table 1.) according to reviewer suggestion.
Comment #6,
In lines 112-113, the abbreviation BBCH needs to be defined.
Answer: We remove BBCH for all body of manuscript (text, figures) and replaced it with more adequate term “vegetation period (days)”.
Comment #7,
Line 133, how was the total protein content determined? Include a brief description.
Answer: We agree; the description of total protein content was added to the manuscript.
...“ Finally, the total protein percentage of nitrogen contained in organic compounds and in ammonia and ammonium in inorganic compounds and the total nitrogen concentration in mg.kg-1 were determined after the mineralisation process with H2SO4; as in Kjeldahl “...
Comment #8,
Lines 165-170, this paragraph must be included in the introduction section.
Answer: See reaction to Comment #3.
Comment #9,
Line 186: “nanoparticles’ input into plants from leaves and the related physiological effects are a bit different in comparison to soluble zinc-based fertilizers [35].” What did the authors want to say exactly when referring to soluble zinc-based fertilizers? Both Zn fertilizers applied foliar and to the soil?
Answer: We agree, information about Zn foliarlly-applied fertilizers vs Zn ionic form was changed in more logical framework.
…” and therefore the nanoparticle input to plants from leaves and the related physiological effects are a little different to those in the zinc-based ionic form Zn(2+) and micro-particle foliarly-applied fertilisers [4]. ”…
Comment #10 + Comment #17;
Table 4. When the parameters were analyzed? At the end of the experiment, thus, 72 days? Instead of referring to “during the growing season” in the Table heading the specific day must be included. + Comment #11, Instead of including the P values I suggest to delete this column and include symbols of asterisks indicating the significance. (e.g *P value <0.05, **P value <0.01.)
Answer: Yes, we accept all of suggestions, we resigned with asterisks Table 4, but also Figure 5.
Comment #12,
The term “qualitative parameters” is confusing. They refers to parameters of quality, but they are quantitative (you have a mean value for each). Revise this term over the manuscript. Moreover, include the organ that was analyzed (grains).
Answer: The term “qualitative parameters” was changed in whole body of manuscript, and we used more appropriate term “nutritional”.
Comment #13,
Lines 200-201: “This was expected as the concentration of Zn was significantly lower than the reported 0.1% value that have the harmful effects when applied foliarly [36].” Include this information in the introduction section.
Answer: This point in discussion section was detailed, and extended about appropriate concentration range.
…” ZnO NPs foliar-applied treatments and the control; but this was expected because the concentration of zinc was significantly lower than the reported 0.1% value that usually has harmful effects when ionic form of Zn(2+) corresponding metal-based fertiliser is applied foliarly [11].
In addition, although the positive effect of approximately 62 mg/L-1 ZnO NPs concentration on growth parameters was confirmed by Singh, (2019) and higher concentration effect was indicated by [7], or [12], the specific dose-dependency of ZnO NPs foliarly-applied nanoparticles appears more complex when linked with factors such as size, shape, surface-to-volume ratio [8],”…
Comment #14,
Lines 203-204: “Thus, the positive effect of ZnO NPs on standard growth parameters is more significant when uptaken by roots even at low concentration range [38].” This sentence is not clear. Are the authors comparing to ZnNPs applied to soil? Or Zn (non-NPs) applied to the soil?
Answer: According to suggestion the Zn-soil-related points was omitted from manuscript.
Comment #15,
Lines 215-216: “ZnNPs also could inhibit the pathogen manifestation, which in turn affected the quality of final crop products [11” Did the authors observe this effect in their experiment? If not, remove this dissertation.
Answer: There was no observation the pathogen manifestation during vegetation period (either ZnO NPs variant or control), and we decide to do not place this information to manuscript.
Comment #16,
Line 220-221: authors may refer to control plants, not to control experiments.
Answer: We agree, this point was changed according to reviewer suggestion:
...„ In contrast, although experimental total protein contents were not statistically different in the control and ZnO NPs treated plants, we identified a statistically significant decrease in starch content in the control plants (Table 4). “...
Comment #17,
Figure 5. Remove “variant” from legend and figure caption. In the X ax include the days (31, 37, etc.) instead of the number of measurements to present a quantitative ax instead of the qualitative ax. Include *, **, etc. with the ANOVA analysis for each day.
Answer: see Comment #10.
References
Sturikova, H.; Krystofova, O.; Huska, D.; Adam, V., Zinc, zinc nanoparticles and plants. Journal of Hazardous Materials 2018, 349, 101-110. Sabir, S.; Arshad, M.; Chaudhari, S. K., Zinc oxide nanoparticles for revolutionizing agriculture: synthesis and applications. The Scientific World Journal 2014, 2014, 1 - 8. Estrada-Urbina, J.; Cruz-Alonso, A.; Santander-González, M.; Méndez-Albores, A.; Vázquez-Durán, A., Nanoscale zinc oxide particles for improving the physiological and sanitary quality of a Mexican landrace of red maize. Nanomaterials 2018, 8, (4), 247. Mousavi Kouhi, S. M.; Lahouti, M.; Ganjeali, A.; Entezari, M. H., Comparative phytotoxicity of ZnO nanoparticles, ZnO microparticles, and Zn2+ on rapeseed (Brassica napus L.): investigating a wide range of concentrations. Toxicological & Environmental Chemistry 2014, 96, (6), 861-868. Wagner, G.; Korenkov, V.; Judy, J.; Bertsch, P., Nanoparticles composed of Zn and ZnO inhibit Peronospora tabacina spore germination in vitro and P. tabacina infectivity on tobacco leaves. Nanomaterials 2016, 6, (3), 50. Gkanatsiou, C.; Ntalli, N.; Menkissoglu-Spiroudi, U.; Dendrinou-Samara, C., Essential metal-based nanoparticles (copper/iron nps) as potent nematicidal agents against Meloidogyne spp. Journal of Nanotechnology Research 2019, 2, 043-057. Prasad, T. N. V. K. V.; Sudhakar, P.; Sreenivasulu, Y.; Latha, P.; Munaswamy, V.; Reddy, K. R.; Sreeprasad, T. S.; Sajanlal, P. R.; Pradeep, T., Effect of nanoscale zinc oxide particles on the germination, growth and yield of peanut. Journal of Plant Nutrition 2012, 35, (6), 905-927. Xiang, L.; Zhao, H.-M.; Li, Y.-W.; Huang, X.-P.; Wu, X.-L.; Zhai, T.; Yuan, Y.; Cai, Q.-Y.; Mo, C.-H., Effects of the size and morphology of zinc oxide nanoparticles on the germination of Chinese cabbage seeds. Environmental Science and Pollution Research 2015, 22, (14), 10452-10462. Tian, B.; Luan, S.; Zhang, L.; Liu, Y.; Zhang, L.; Li, H., Penalties in yield and yield associated traits caused by stem lodging at different developmental stages in summer and spring foxtail millet cultivars. Field Crops Research 2018, 217, 104-112. Duflo, E.; Banerjee, A., Handbook of field experiments. first edition ed.; Elsevier: 2017; Vol. 1. Drissi, S.; Houssa, A. A.; Bamouh, A.; Benbella, M., Corn silage (Zea mays L.) response to zinc foliar spray concentration when grown on sandy soil. Journal of Agricultural Science 2015, 7, (2), 68 - 79. Torabian, S.; Zahedi, M.; Khoshgoftar, A. H., Effects of foliar spray of two kinds of zinc oxide on the growth and ion concentration of sunflower cultivars under salt stress. Journal of plant nutrition 2016, 39, (2), 172-180.

Reviewer 2 Report
Dear Authors
this is an interesting article on the use of zinc oxide nanoparticles on foxtail millet. Over all, the experimental protocols are well established and correctly analysed. The results are soundly presented. I would suggest only some language editing before publishing and some minor changes according to the following comments:
Please explain the test concentration of the formulate used under field conditions. On what bases did authors choose to test this specific concentration? Since authors used an adjuvant to facilitate the penetration, should there be in the experimental protocol also a "blank treatment" with just this adjuvant? Do authors have contacted maybe preliminary trials to sustain no effect of such a treatment? Please consider in the reference list the work entitled "Essential Metal-Based Nanoparticles (Copper/Iron NPs) as Potent Nematicidal Agents against Meloidogyne spp. J Nanotechnol Res 2019; 1 (2): 044-058"
Author Response
Reviewer II
Comment #1, + Reviewer 1 (Comment #1),
I would suggest only some language editing before publishing and some minor changes according to the following comments:
Answer: English language corrections of manuscript were checked by English lector Dr. Raymond Joseph Marshall.
Comment #2,
The test concentration of the formulate used under field conditions. On what bases did authors choose to test this specific concentration?
Answer: This information was included into manuscript.
…”In addition, although the positive effect of approximately 62 mg/L-1 ZnO NPs concentration on growth parameters was confirmed by Singh, (2019) and higher concentration effect was indicated by [1], or [2], the specific dose-dependency of ZnO NPs foliarly-applied nanoparticles appears more complex when linked with factors such as size, shape, surface-to-volume ratio [3],”…
Comment #3,
Since authors used an adjuvant to facilitate the penetration, should there be in the experimental protocol also a "blank treatment" with just this adjuvant? Do authors have contacted maybe preliminary trials to sustain no effect of such a treatment?
Answer: The point with “adjuvant” according to reviewer suggestion was added into manuscript.
… “The ZnO NPs were then suspended in a solution of adjuvant SILWET STAR® before foliar application because this facilitates penetration of nanoparticles across wax sub-structures [4, 5] and also used for control as a “blank” treatment lacking ZnO NPs.”…
Comment #4,
Please consider in the reference list the work entitled "Essential Metal-Based Nanoparticles (Copper/Iron NPs) as Potent Nematicidal Agents against Meloidogyne spp. J Nanotechnol Res 2019; 1 (2): 044-058"
Answer: The reference was considered and added into manuscript.
…” The positive effects of metal (metal oxides) nanoparticles (NPs) on crops have recently been recorded, and these include being an effective growth stimulator in dry seasons, enhancing the yield and nutritional quality of agricultural products [1, 6, 7] and providing effective mediation as an active anti-pathogen agent [8-10].“...
References
Prasad, T. N. V. K. V.; Sudhakar, P.; Sreenivasulu, Y.; Latha, P.; Munaswamy, V.; Reddy, K. R.; Sreeprasad, T. S.; Sajanlal, P. R.; Pradeep, T., Effect of nanoscale zinc oxide particles on the germination, growth and yield of peanut. Journal of Plant Nutrition 2012, 35, (6), 905-927. Torabian, S.; Zahedi, M.; Khoshgoftar, A. H., Effects of foliar spray of two kinds of zinc oxide on the growth and ion concentration of sunflower cultivars under salt stress. Journal of plant nutrition 2016, 39, (2), 172-180. Xiang, L.; Zhao, H.-M.; Li, Y.-W.; Huang, X.-P.; Wu, X.-L.; Zhai, T.; Yuan, Y.; Cai, Q.-Y.; Mo, C.-H., Effects of the size and morphology of zinc oxide nanoparticles on the germination of Chinese cabbage seeds. Environmental Science and Pollution Research 2015, 22, (14), 10452-10462. Burghardt, M.; Schreiber, L.; Riederer, M., Enhancement of the diffusion of active ingredients in barley leaf cuticular wax by monodisperse alcohol ethoxylates. Journal of Agricultural and Food Chemistry 1998, 46, (4), 1593-1602. Räsch, A.; Hunsche, M.; Mail, M.; Burkhardt, J.; Noga, G.; Pariyar, S., Agricultural adjuvants may impair leaf transpiration and photosynthetic activity. Plant Physiology and Biochemistry 2018, 132, 229-237. López-Vargas, E.; Ortega-Ortíz, H.; Cadenas-Pliego, G.; de Alba Romenus, K.; Cabrera de la Fuente, M.; Benavides-Mendoza, A.; Juárez-Maldonado, A., Foliar application of copper nanoparticles increases the fruit quality and the content of bioactive compounds in tomatoes. Applied Sciences 2018, 8, (7), 1020. Yang, F.; Hong, F.; You, W.; Liu, C.; Gao, F.; Wu, C.; Yang, P., Influence of nano-anatase TiO2 on the nitrogen metabolism of growing spinach. Biological Trace Element Research 2006, 110, (2), 179-190. Bellesi, F. J.; Arata, A. F.; Martínez, M.; Arrigoni, A. C.; Stenglein, S. A.; Dinolfo, M. I., Degradation of gluten proteins by Fusarium species and their impact on the grain quality of bread wheat. Journal of Stored Products Research 2019, 83, 1-8. Wagner, G.; Korenkov, V.; Judy, J.; Bertsch, P., Nanoparticles composed of Zn and ZnO inhibit Peronospora tabacina spore germination in vitro and P. tabacina infectivity on tobacco leaves. Nanomaterials 2016, 6, (3), 50. Gkanatsiou, C.; Ntalli, N.; Menkissoglu-Spiroudi, U.; Dendrinou-Samara, C., Essential metal-based nanoparticles (copper/iron nps) as potent nematicidal agents against Meloidogyne spp. Journal of Nanotechnology Research 2019, 2, 043-057.
